# Combining Reconstruction-based Unsupervised Anomaly Detection with Supervised Segmentation for Brain MRIs

**Finn Behrendt**[1]                                                     FINN.BEHRENDT@TUHH.DE
**Debayan Bhattacharya**[1]                               DEBAYAN.BHATTACHARYA@TUHH.DE
**Lennart Maack** [1]                                              LENNART.MAACK@TUHH.DE
**Julia Krüger**[2]                                      JULIA.KRUEGER@JUNG-DIAGNOSTICS.DE
**Roland Opfer**[2]                                       ROLAND.OPFER@JUNG-DIAGNOSTICS.DE
**Alexander Schlaefer**[1]                                             SCHLAEFER@TUHH.DE

[1] *Institute of Medical Technology and Intelligent Systems, Hamburg University of Technology, Hamburg, Germany*

[2] *Jung Diagnostics GmbH, Hamburg, Germany*

## Abstract

In contrast to supervised deep learning approaches, unsupervised anomaly detection (UAD) methods can be trained with healthy data only and do not require pixel-level annotations, enabling the identification of unseen pathologies. While this is promising for clinical screening tasks, reconstruction-based UAD methods fall short in segmentation accuracy compared to supervised models. Therefore, self-supervised UAD approaches have been proposed to improve segmentation accuracy. Typically, synthetic anomalies are used to train a segmentation network in a supervised fashion. However, this approach does not effectively generalize to real pathologies. We propose a framework combining reconstruction-based and self-supervised UAD methods to improve both segmentation performance for known anomalies and generalization to unknown pathologies. The framework includes an unsupervised diffusion model trained on healthy data to produce pseudo-healthy reconstructions and a supervised Unet trained to delineate anomalies from deviations between input-reconstruction pairs. Besides the effective use of synthetic training data, this framework allows for weakly-supervised training with small annotated data sets, generalizing to unseen pathologies. Our results show that with our approach, utilizing annotated data sets during training can substantially improve the segmentation performance for in-domain data while maintaining the generalizability of reconstruction-based approaches to pathologies unseen during training.

**Keywords:** Unsupervised Anomaly Detection, Diffusion Models, Brain MRI, Self Supervision, Weak Supervision

## 1. Introduction

Deep learning (DL) methods have advanced in their ability to detect and segment brain pathologies in MRI images (Lundervold and Lundervold, 2019). However, acquiring annotated data for each pathology is a challenge, especially when considering screening tasks, where the objective is to detect any potential anomaly.
Unsupervised anomaly detection (UAD) provides a potential solution by modeling the distribution of healthy brain MRI scans to identify anomalies as outliers. A common technique in UAD is reconstruction-based anomaly detection, where generative models (GM) are trained to reconstruct healthy brain images. At test time, the GMs fail to replicate

pathologies, thereby revealing anomalies through discrepancies between input and reconstruction. This method only necessitates healthy data and enables the identification of pathologies not encountered during training, which poses a challenge for supervised models. However, the performance of reconstruction-based UAD methods is often surpassed by supervised models when sufficient task-specific data is available (Chen et al., 2020; Baur et al., 2021b). Unlike supervised methods, UAD methods that rely on reconstructions do not directly learn the relationship between abnormal patterns and their corresponding annotations. Instead, the segmentation map is a byproduct of measuring the discrepancy between input and reconstruction. This results in a noisy anomaly map with potential false positives caused by the GM's imperfect reconstructions. Consequently, distinguishing actual anomalies from normal reconstruction errors can be challenging. An alternative approach is self-supervised UAD, where synthetic anomalies are introduced to the healthy brain images to train a segmentation network in a supervised manner. Unlike reconstruction-based UAD, this strategy produces distinct anomaly maps with high specificity, simplifying the discrimination of abnormal structures similar to the synthesized anomalies. However, the segmentation performance depends on the nature of the generated anomalies and tends to have limited generalization to real pathologies (Lagogiannis et al., 2023; Cai et al., 2023). In this study, we aim to combine the strong generalization capabilities and high sensitivity of reconstruction-based methods with the high specificity of self-supervised methods. We develop a framework that employs a denoising diffusion probabilistic model (DDPM; DM) to generate pseudo-healthy reconstructions of potentially abnormal input images (reconstruction branch). Furthermore, an Unet is trained to segment anomalies based on the residual of the input and the pseudo-healthy reconstruction (segmentation branch). We consider different settings to obtain the annotations for the supervised training of the Unet. First, in the self-supervised setting, we introduce synthetically generated anomalies to healthy brain MRIs. Second, in the semi-supervised setting, we utilize a small amount of annotated data containing real pathologies. At test time, the unsupervised anomaly maps from the reconstruction branch and the supervised predictions from the segmentation branch are fused to a final anomaly score.

The results demonstrate that in contrast to self-supervised methods, our approach allows to integrate supervision while maintaining the generalizability of the underlying reconstruction branch. Specifically, we can improve the Dice score of reconstruction-based UAD methods from 58.55 % to 69.68 % for tumors when using the same pathologies for training, while the Dice score for stroke lesions unseen during training increases from 24.74 % to 26.77 %.

## 2. Related Work

For reconstruction-based UAD, different architectures have been proposed as GM. While the majority focuses on Autoencoders (AE) (Baur et al., 2021a) or Variational autoencoders (VAE) (Zimmerer et al., 2019; Chen et al., 2020; Bercea et al., 2023a,c), also vector-quantized VAEs (Pinaya et al., 2022) and GANs (Nguyen et al., 2021) have been employed. Moreover, it has been shown that utilizing denoising tasks for regularization with Unet-like AEs can improve the UAD performance (Kascenas et al., 2022, 2023). Consequently, DDPMs have emerged as a GM for reconstruction-based UAD (Wyatt et al., 2022; Behrendt et al., 2023a,b; Bercea et al., 2023b). In self-supervised UAD, typically, synthetic anomalies

are incorporated into normal brain images. Subsequently, Unets are trained to segment these synthetic anomalies (Tan et al., 2021, 2022; Cho et al., 2022; Meissen et al., 2022a). We note that while AE-based reconstruction methods may also fall under the category of self-supervised techniques, within this work, the term "self-supervised" refers to the aforementioned approach of training segmentation models using synthetic anomalies. Expanding on this strategy, DRAEM (Zavrtanik et al., 2021) employs a dual-network architecture comprising a generator and a segmentation network. The generator is trained to eliminate synthetic anomalies, thereby providing a pseudo-healthy reconstruction. The segmentation network is then used to segment the generated anomalies, given the concatenation of abnormal input and pseudo-healthy reconstruction. Note that for the generator network in DRAEM, inpainting of synthetic anomalies is enforced by calculating the reconstruction loss between reconstruction and the anomaly-free input. In contrast, in our approach, the reconstruction model is trained on healthy data in an unsupervised fashion to remove any abnormal structure that is not part of the healthy training distribution. Hence, we expect this approach to generalize more readily to real pathologies. The authors (Liu et al., 2022) take a similar approach, aiming to improve supervised segmentation performance by augmenting a dual-branch Unet with pseudo-healthy reconstructions. These reconstructions are generated by a Soft-Intro VAE trained on healthy data. In contrast, our proposed framework does not solely depend on supervised predictions. Instead, these predictions are combined with the unsupervised anomaly scores derived from reconstructions of a DM. We hypothesize that this combination enables general anomaly detection, particularly for pathologies unseen during training.

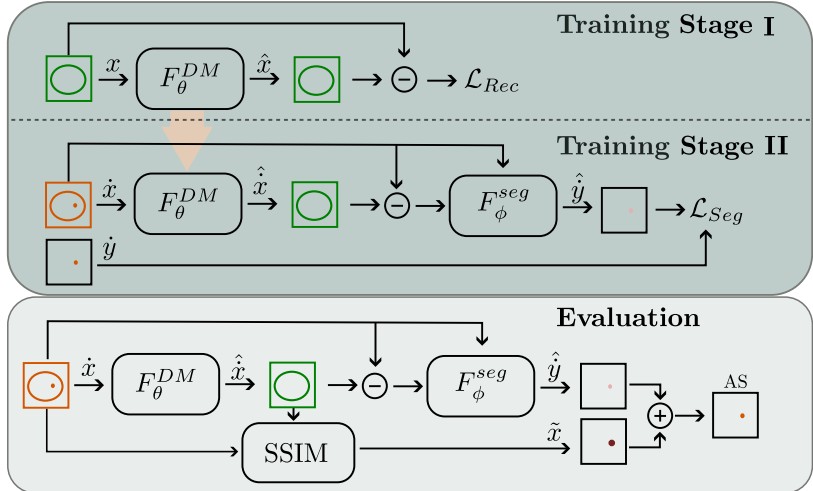

Figure 1: Schematic drawing of SADM. In Stage I, $F_\theta^{DM}$ is trained to reconstruct healthy brain images. In stage II, the parameters $\theta$ are fixed, and the segmentation network $F_\phi^{seg}$ is trained, either on synthetic anomalies (self-supervised) or real pathologies (semi-supervised). At test time, the supervised prediction $\hat{y}$ and the unsupervised anomaly map $\tilde{x}$ are combined to the final anomaly score (AS).

## 3. Method

In this section, we introduce our framework for supervised anomaly detection with DMs (SADM), detailed schematically in Figure 1.

### 3.1. Supervised Anomaly Detection with Diffusion Models (SADM)

SADM integrates two primary branches: a DM for generating pseudo-healthy reconstructions (reconstruction branch) and a supervised Unet for segmentation (segmentation branch). We train SADM in two sequential stages.

#### STAGE I: UNSUPERVISED RECONSTRUCTION

In the first stage, our objective is to train the DM to reconstruct healthy brain scans $\hat{x} = F_\theta^{DM}(x)$ where $x \in \mathbb{R}^{H \times W}$. The training of the DM focuses on optimizing parameters $\theta$ to minimize the $l1$-reconstruction loss:

$$\mathcal{L}_{Rec} = |x - \hat{x}|. \tag{1}$$

#### STAGE II: SUPERVISED SEGMENTATION

In the second stage, the pseudo-healthy reconstruction generated by the DM trained in Stage I is utilized to support anomaly segmentation. Given an input scan with a real or synthetic anomaly $\dot{x} \in \mathbb{R}^{H \times W}$ and its corresponding ground truth annotation $\dot{y} \in \mathbb{R}^{H \times W}$, we use the DM, trained in Stage I, to generate the pseudo-healthy reconstruction $\hat{\dot{x}} = F_\theta^{DM}(\dot{x})$. Next, we feed both the residual $(\dot{x} - \hat{\dot{x}})$ and the original input $\dot{x}$ into a Unet. After encoding both inputs, the resulting features are concatenated at each layer and fed to the Unet decoder to predict the segmentation map $\hat{\dot{y}} = F_\phi^{seg}(\dot{x} - \hat{\dot{x}}, \dot{x})$. Only the Unet parameters $\phi$ are optimized to minimize the cross-entropy (CE) segmentation loss during Stage II

$$\mathcal{L}_{Seg} = CE(\hat{\dot{y}}, \dot{y}). \tag{2}$$

#### ANOMALY DETECTION

The anomaly detection process leverages both components of our framework for anomaly segmentation. Given a potentially abnormal input $\dot{x}$, we generate a reconstruction $\hat{\dot{x}} = F_\theta^{DM}(\dot{x})$ by the DM. Next, we utilize $F_\phi^{seg}$ to derive the supervised anomaly prediction $\hat{\dot{y}} = F_\phi^{seg}(\dot{x} - \hat{\dot{x}}, \dot{x})$. In addition, we utilize the pixel-wise structural similarity (SSIM (Wang et al., 2004)) between input and reconstruction $\tilde{x} = 1 - SSIM(\dot{x} - \hat{\dot{x}})$ for unsupervised anomaly scoring. The anomaly score (AS) is then derived as a combination of the unsupervised anomaly map and supervised anomaly prediction

$$\text{Anomaly Score (AS)} = \tilde{x} + \hat{\dot{y}}. \tag{3}$$

For pathologies similar to the anomalies seen during training, the supervised anomaly prediction will feature higher probabilities in abnormal regions, refining the unsupervised anomaly map. For unseen pathologies, the predicted probabilities are low such that $\tilde{x}$ is unaltered. We hypothesize that this combination allows for comprehensive anomaly detection, leveraging the unsupervised anomaly map for general anomaly identification and the supervised prediction for precise segmentation of known abnormal patterns.

## 4. Experimental Setup

### 4.1. Data

We use T1-weighted MRIs from the IXI data set to train the DM in Stage I. We separate a healthy test set consisting of 160 samples. The remaining data is partitioned into five training sets (N=358) and validation sets (N=44) for cross-validation. In Stage II, we utilize the strategy applied in (Zavrtanik et al., 2021) to generate pairs of synthetic anomalies and ground truth annotation based on the IXI data set (DRAEM). Additional information about the generation process and exemplary anomalies are provided in Appendix C. Additionally, for the weakly supervised setting, we utilize small subsets containing approximately 10% of the BraTS21 (BRATS, N=1251) (Baid et al., 2021; Bakas et al., 2017; Menze et al., 2014), and ATLAS-v2.0 (ATLAS, N=655) (Liew et al., 2022) data sets. For evaluation, we utilize the remaining 1151 and 589 samples of the BRATS and ATLAS data sets, respectively. Furthermore, we utilize the augmented IXI test set (DRAEM) to assess the segmentation performance concerning synthetic anomalies.

**Pre- and post-processing:** We resample all T1 MRI scans to a resolution of $[1 \times 1 \times 1]$ mm and register them to the SRI24-Atlas (Rohlfing et al., 2010). Subsequently, we perform skull-stripping using HD-BET (Isensee et al., 2019) leading to volumes of size $[192 \times 192 \times 160]$ voxels. Finally, we apply bias-field corrections, reduce the resolution by a factor of two and crop 15 top and bottom slices in the transverse plane. For post-processing, we apply median filtering with a kernel size of 5 to the unsupervised anomaly maps.

### 4.2. Implementation Details

We utilize DMs as GM within our proposed framework to generate pseudo-healthy reconstructions[1]. Specifically, we use conditioned DDPMs (cDDPM) following the implementation of (Behrendt et al., 2023b). For the supervised segmentation of the residual image, we utilize a Unet (Ronneberger et al., 2015) like architecture, adapted from (Kascenas et al., 2022). The volumes are processed in a slice-wise fashion, sampling slices uniformly during training. At test time, we reconstruct the full volume by iterating over all slices. We compare our framework against different established baselines. We compare reconstruction-based AEs and VAEs (Baur et al., 2021a), FAEs (Meissen et al., 2022b), DDPMs (Wyatt et al., 2022), pDDPMs (Behrendt et al., 2023a) and cDDPMs (Behrendt et al., 2023b). Furthermore, we compare the feature-based reverse distillation method (RD) (Deng and Li, 2022), the self-supervised Poisson image interpolation (PII) (Tan et al., 2021) and DRAEM-Net (Zavrtanik et al., 2021) approaches. Note that for PII we perform the anomaly generation based on the IXI data set. For all reconstruction-based methods, we utilize SSIM for anomaly scoring with a Gaussian kernel with standard deviation of $\sigma_{ssim} = 1$, leading to a window size of $k_{ssim} = 9$. Implementation details of our proposed framework and compared baselines are provided in Appendix B.

---

1. Code available at
   https://github.com/FinnBehrendt/Supervised-Anomaly-Detection-with-Diffusion-Models

| | Model | Training Data | | Test Data | | | | | |
|---|---|---|---|---|---|---|---|---|---|
| | | | | **BRATS** (real) | | **ATLAS** (real) | | **DRAEM** (synthetic) | |
| | | $\mathcal{D}_{healthy}$ | $\mathcal{D}_{unhealthy}$ | $\lceil$DICE$\rceil$ | AUPRC | $\lceil$DICE$\rceil$ | AUPRC | $\lceil$DICE$\rceil$ | AUPRC |
| **I. Unsupervised** | AE | IXI | None | 39.16 ± 0.64 | 35.95 ± 0.70 | 14.14 ± 0.28 | 11.84 ± 0.37 | 9.91 ± 0.04 | 5.27 ± 0.04 |
| | VAE | IXI | None | 39.25 ± 0.50 | 36.07 ± 0.56 | 14.52 ± 0.37 | 12.18 ± 0.39 | 9.83 ± 0.14 | 5.28 ± 0.08 |
| | DAE | IXI | None | 55.93 ± 0.66 | 56.42 ± 0.84 | 19.95 ± 0.96 | 18.18 ± 0.98 | 12.50 ± 0.31 | 7.50 ± 0.22 |
| | FAE | IXI | None | 43.04 ± 0.49 | 42.04 ± 0.41 | 17.59 ± 0.15 | 13.91 ± 0.10 | **19.60 ± 0.49** | **13.68 ± 0.25** |
| | RD | IXI | None | 32.90 ± 0.65 | 28.31 ± 0.86 | 19.45 ± 0.25 | 15.51 ± 0.20 | 19.55 ± 0.60 | 13.17 ± 0.61 |
| | DDPM | IXI | None | 48.65 ± 0.90 | 46.93 ± 1.02 | 17.86 ± 0.87 | 14.70 ± 0.70 | 10.37 ± 0.23 | 6.04 ± 0.27 |
| | pDDPM | IXI | None | 55.93 ± 0.28 | 55.44 ± 0.36 | 21.79 ± 0.40 | 19.12 ± 0.43 | 14.59 ± 0.47 | 9.27 ± 0.31 |
| | cDDPM | IXI | None | **58.55 ± 0.78** | **59.09 ± 0.91** | **24.74 ± 1.15** | **21.76 ± 0.98** | 11.94 ± 0.52 | 7.31 ± 0.43 |
| **II. Self-Supervised** | PII | None | PII | 30.38 ± 2.46 | 24.66 ± 2.54 | 9.81 ± 1.93 | 7.31 ± 1.64 | 23.44 ± 1.61 | 15.09 ± 0.97 |
| | DRAEM-Net | None | DRAEM | 24.78 ± 4.21 | 18.49 ± 4.05 | 12.65 ± 1.90 | 9.51 ± 1.75 | **79.77 ± 2.37** | **83.39 ± 2.34** |
| | Unet | None | DRAEM | 40.75 ± 3.30 | 37.64 ± 3.92 | 16.91 ± 0.38 | 15.25 ± 0.26 | 76.03 ± 1.21 | 80.30 ± 1.32 |
| | Unet$_{res}$ | IXI | DRAEM | 45.80 ± 3.22 | 44.05 ± 4.09 | 18.44 ± 0.47 | 16.81 ± 0.44 | 77.43 ± 1.16 | 81.93 ± 1.23 |
| | SADM | IXI | DRAEM | 50.81 ± 0.57 | 49.81 ± 0.81 | 23.82 ± 0.32 | 20.71 ± 0.35 | 73.77 ± 2.50 | 71.85 ± 3.02 |
| | SADM$_{res}$ | IXI | DRAEM | **60.53 ± 0.54** | **60.27 ± 1.02** | **27.78 ± 0.14** | **24.57 ± 0.13** | 76.72 ± 1.30 | 75.45 ± 1.96 |
| **III. Weakly-Supervised** | Unet | None | BRATS | 64.81 ± 0.21 | 69.24 ± 0.33 | 11.82 ± 0.60 | 10.32 ± 0.61 | **24.83 ± 1.10** | **20.96 ± 1.46** |
| | Unet$_{res}$ | IXI | BRATS | 67.01 ± 0.70 | 71.80 ± 0.87 | 17.33 ± 1.31 | 15.55 ± 1.50 | 19.93 ± 2.40 | 16.41 ± 2.64 |
| | SADM | IXI | BRATS | 69.01 ± 0.21 | 72.62 ± 0.46 | 25.25 ± 0.58 | 21.03 ± 0.50 | 14.93 ± 0.51 | 11.65 ± 0.66 |
| | SADM$_{res}$ | IXI | BRATS | **69.68 ± 0.48** | **73.34 ± 0.85** | **26.77 ± 0.65** | **23.22 ± 0.86** | 17.11 ± 1.78 | 14.47 ± 1.91 |
| | Unet | None | ATLAS | 35.13 ± 2.97 | 32.87 ± 3.07 | 46.30 ± 0.72 | 46.37 ± 0.73 | **29.11 ± 1.02** | **24.55 ± 1.91** |
| | Unet$_{res}$ | IXI | ATLAS | 36.82 ± 4.18 | 34.91 ± 4.92 | 47.36 ± 0.80 | **47.61 ± 0.88** | 22.07 ± 2.20 | 17.94 ± 2.39 |
| | SADM | IXI | ATLAS | 58.52 ± 0.60 | 57.17 ± 1.60 | 46.40 ± 0.17 | 44.71 ± 0.15 | 16.10 ± 1.10 | 12.81 ± 1.09 |
| | SADM$_{res}$ | IXI | ATLAS | **58.85 ± 0.44** | **57.68 ± 1.23** | **47.64 ± 1.40** | 46.13 ± 1.36 | 17.77 ± 1.82 | 14.49 ± 1.73 |

Table 1: Segmentation performance regarding DICE and AUPRC. **Block I:** Unsupervised approaches, trained with healthy data. **Block II:**, Self-supervised approaches, trained with synthetic anomalies. **Block III:** Weakly-supervised approaches, trained with real pathologies. $\mathcal{D}_{healthy}$ and $\mathcal{D}_{unhealthy}$ represent the type of data used during training.

## 5. Experiments

For all our experiments, we evaluate the BRATS and ATLAS data sets containing real pathologies and the IXI data set augmented with synthetic anomalies (DRAEM). We report the mean ± standard deviation across the different folds for the best possible Dice Score ([DICE]) as well as the Area under Precision-Recall Curve (AUPRC) to assess the segmentation performance. We evaluate different variants of SADM. In SADM$_{res}$, the residual of input and reconstruction and the (abnormal) input are fed to the Unet, whereas in SADM, only the input is used. Furthermore, we consider Unet and Unet$_{res}$, where, in contrast to SADM only the prediction of the Unet is used, ignoring the anomaly map of the unsupervised reconstruction branch. In Appendix D, we provide an ablation study on the weighted combination of the segmentation and reconstruction branch.

### 5.1. Training with Synthetic Anomalies

We evaluate our approach in different settings. First, we assume the typical UAD case where only data with healthy labels is available. We use synthetic anomalies to obtain a supervised signal for the segmentation branch in SADM. We utilize the generation process proposed in DRAEM (Zavrtanik et al., 2021) to generate the anomalies. In this setting, we compare our framework to various UAD baselines. Results are reported in block I and block II of Table 1. Across the compared UAD baselines in block I, cDDPMs show the highest

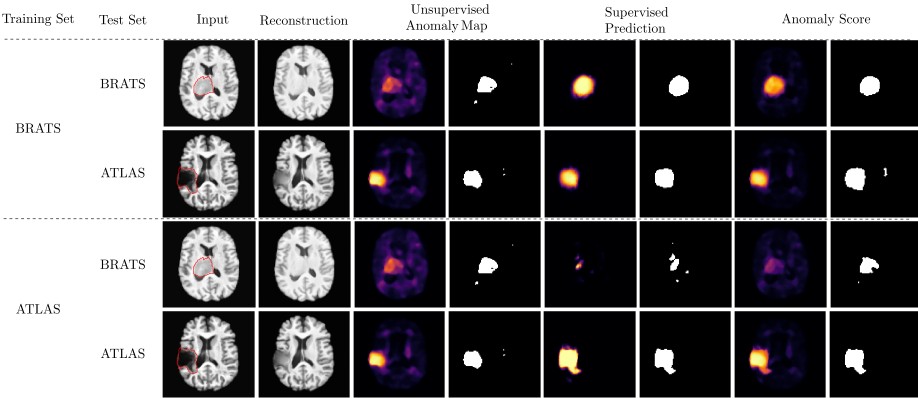

Figure 2: Examplary test cases for SADM$_{res}$, trained and evaluated in the weakly-supervised setting with the BRATS and ATLAS data sets, respectively. For visualization purposes, we provide exemplary binary segmentation maps for the unsupervised anomaly score, the supervised prediction and the final AS, respectively. We derive the binarization threshold by optimizing for the best possible dice score.

segmentation performance for real pathologies. Hence, we consider them as a reconstruction model for the SADM framework. For real pathologies, SADM$_{res}$ outperforms cDDPMs with performance improvements of 3.4 %, 12.3 % for the BRATS and ATLAS data sets, respectively. Considering the synthetic anomalies in the DRAEM data set, a substantially higher DICE of 76.72 % is reported for SADM$_{res}$ compared to the DICE of 11.94 % achieved by cDDPMs. Notably, while the DRAEM-Net shows relative performance improvements of 10.5 % over SADM$_{res}$ for synthetic anomalies, it fails to generalize to the real pathologies in the BRATS and ATLAS data sets. Even the Unet, trained with the same synthetic anomalies as in DRAEM-Net, outperforms DRAEM-Net considering real pathologies.

Comparing SADM and SADM$_{res}$, we observe that utilizing the residual of abnormal input and pseudo-healthy reconstruction in addition to the abnormal input substantially improves the segmentation performance across all data sets.

## 5.2. Training with Real Pathologies

In this section we investigate using our framework in a weakly-supervised setting. Instead of generating synthetic anomalies, we assume a small amount of annotated data is available and consider a subset of the BRATS and ATLAS data sets for training, respectively. We only train with one data set at a time to evaluate the generalization to unseen pathologies. The results for this weakly-supervised setting are reported in block III of Table 1. Using a small subset of annotated data substantially improves the segmentation of all models when evaluating the same (in-domain) data set. However, the segmentation performance of Unet and Unet$_{res}$ is poor for data sets containing pathologies unseen during training. In contrast, both SADM and SADM$_{res}$ enhance the segmentation performance on in-domain data while maintaining or even improving the performance of unsupervised cDDPMs for

unseen pathologies. A visualization of the anomaly maps coming from different branches of the SADM framework is provided in Figure 2.

## 6. Discussion and Conclusion

A significant challenge of supervised methods that UAD addresses is the need for annotated training data. This is especially crucial when considering screening tasks where the type and shape of potential lesions are unknown. Therefore, it is highly desirable to achieve generalization to different kinds of lesions while minimizing false positive predictions. In this work, we aim for a framework that benefits from the robust generalization of reconstruction-based UAD methods and the high discriminative power of supervised strategies.

Comparing the unsupervised and self-supervised approaches in Table 1, the additional shape information typically improves the segmentation performance with the magnitude of improvement dependent on the lesion type. However, considering purely self-supervised models, it is evident that supervised training based on synthetic data can result in overfitting. In contrast, our proposed framework, improves the segmentation performance for anomalies of known shape and appearance while maintaining or even improving the generalization of reconstruction-based UAD for pathologies unseen during training. This indicates that the framework effectively utilizes the complementary information of the reconstruction and segmentation branches, as highlighted in Figure 2. On the one hand, the supervised segmentation branch enhances the specificity for pathologies similar to the anomalies seen during training. On the other hand, the reconstruction branch maintains the high sensitivity of reconstruction-based UAD for any abnormal pattern unseen during the training of the DM. Furthermore, feeding the residual of input and reconstruction to the Unet in addition to the abnormal input can enhance the segmentation performance, particularly in the self-supervised setting. This indicates that the additional information in the residual may contribute to learning the deviation from a normal representation, potentially reducing the risk of overfitting to specific anomaly shapes. While the DRAEM-Net shares some similarities with our approach, there are significant differences. First, DRAEM-Net uses a generator network trained to remove synthesized anomalies. In contrast, our reconstruction branch employs a DM trained to reconstruct healthy data without explicitly enforcing the removal of specific anomalies. Second, instead of solely relying on the segmentation branch, we combine the supervised prediction with the unsupervised anomaly map derived from the reconstruction branch. As demonstrated in our experiments, these adaptations lead to improved segmentation performance and generalization, enabling the effective use of SADM in a weakly-supervised setting. Therefore, our framework adds a significant feature to UAD approaches, especially considering that some annotated data is typically available.

In summary, our approach shows encouraging results, paving the way for a practical solution for UAD in brain MRI. Limitations are seen in the potential reconstruction of unhealthy structures by the reconstruction branch and in the investigated synthetic anomalies intended initially for industrial defect detection. Despite the demonstrated improvement in performance, we anticipate further enhancements when integrating more realistic synthetic anomalies. Additionally, we intend to include data sets featuring subtler anomalies or different imaging modalities to broaden the evaluation of our approach.

## Acknowledgments

This work was partially funded by grant number KK5208102HV3 and ZF4026303TS9 (Zentrales Innovationsprogramm Mittelstand) and by the Free and Hanseatic City of Hamburg (Interdisciplinary Graduate School).

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

## Appendix A. Qualitative Comparison

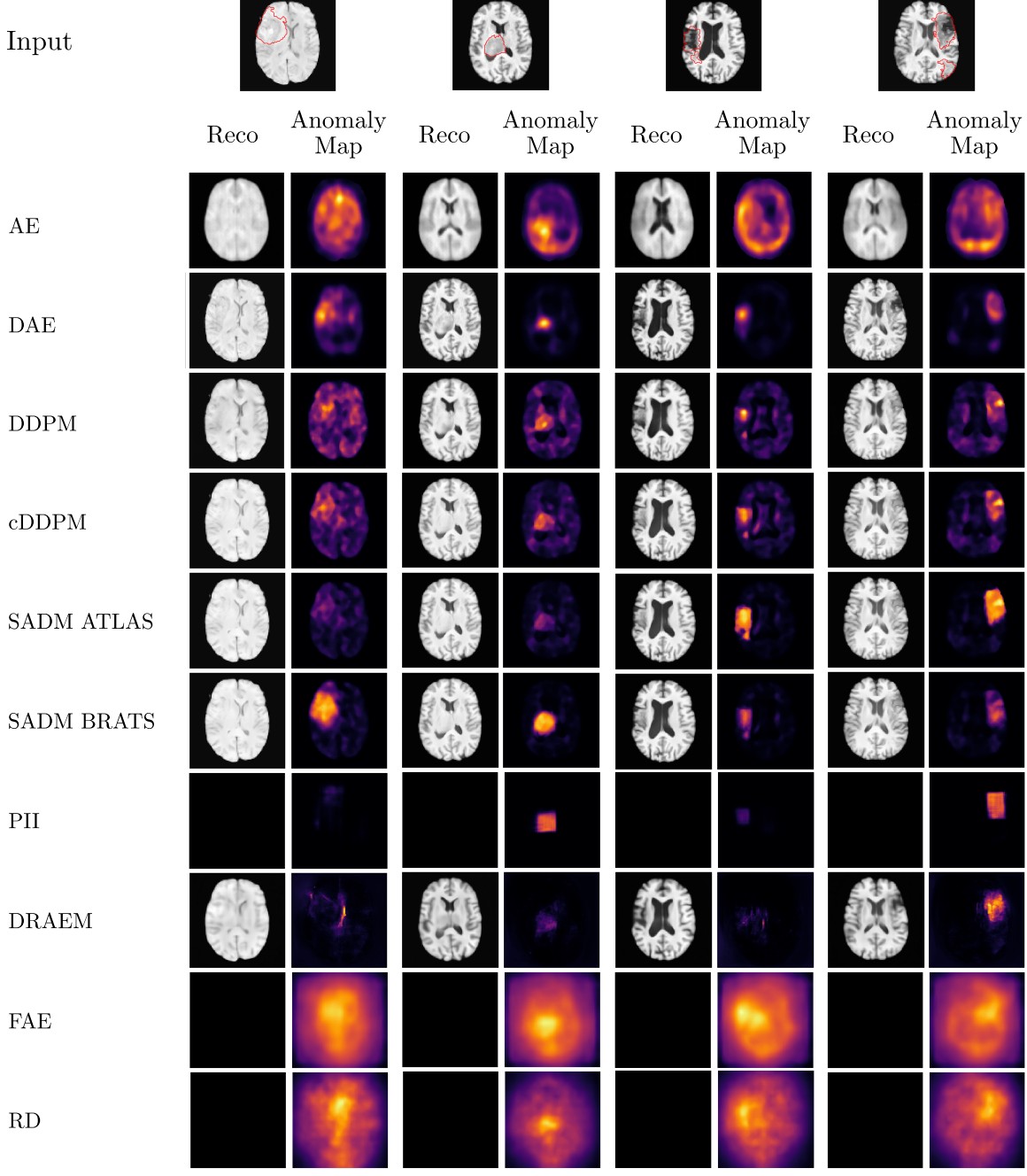

Figure 3: Comparison of baseline models for pathologies from the BRATS (left two columns) and ATLAS (right two columns) data sets.

## Appendix B. Implementation Details

All models are implemented in Pytorch (v0.10). For data handling and augmentation, torchio (Pérez-García et al., 2021) is utilized. We choose the best-performing model checkpoint, measured by the validation set performance. We utilize Adam as an optimizer with a batch size of 32. For data augmentation, we utilize random -blur, -bias, -gamma and -ghosting. All Baselines are implemented following the official GitHub repositories. We train our models on NVIDIA RTX 3090 and V100 GPUs.

### B.1. SADM

Our SADM framework consists of a reconstruction branch and a segmentation branch. In the reconstruction branch, we utilize cDDPMs (Behrendt et al., 2023b) as a generative model. We follow the official implementation[2] and utilize a 3-layer Unet with channel dimensions [128, 128, 256] as a denoising network with a pre-trained resnet50 encoder for conditioning. During training, we uniformly sample noise levels $t \in [0, T]$. At test time, we derive the final reconstruction as an average from reconstructions of different noise levels $t_{test} \in [250, 500, 750]$. For the segmentation branch, we adapt the Unet architecture as employed by (Kascenas et al., 2022). Our base Unet architecture consists of three layers with channel dimensions of 64, 128, and 256, respectively, incorporating group normalization and SiLU activation functions. For $SADM_{res}$, we utilize the same encoder to separately encode the residual of the input and reconstruction, as well as the input itself. The resulting feature maps are then concatenated along the channel dimension at each layer and passed to the decoder, effectively doubling the channel dimensions. A sigmoid layer is added after the final convolution to produce the segmentation output. In stage I and II, we train for 1600 and 600 epochs, with learning rates of 1e-4 and 5e-5, respectively.

### B.2. Baselines

We implement various baseline methods based on the official code with individual adaptations of hyper-parameters that have been shown to improve training stability or performance regarding the validation data. Unless stated otherwise, all models are trained for 1600 epochs, choosing the best checkpoint based on the validation set performance, using Adam as an optimizer. For AEs and VAEs, we use a latent dimension of 128 and set the learning rate to 1e-4. For VAEs, we set $\beta_{KLD} = 0.001$. For RD and DRAEM, we set the learning rate to 1e-4. The DDPM, pDDPM and cDDPM baselines are trained with simplex noise as proposed in (Wyatt et al., 2022) and a learning rate of 1e-5, respectively. Note that for all DDPM-based baselines, we utilize the averaged reconstruction from three different noise levels $t_{test} \in [250, 500, 750]$.

## Appendix C. Synthetic Anomalies

We generate the synthetic anomalies by following the procedure of (Zavrtanik et al., 2021). First, a noise image is generated using Perlin noise (Perlin, 1985), capturing a wide variety of shapes. Subsequently, the noise image is binarized by a uniformly sampled threshold,

---

2. https://github.com/FinnBehrendt/Conditioned-Diffusion-Models-UAD

resulting in an anomaly map $M_a$, that is used as ground truth annotation. The binary map is further processed by three random augmentation functions, sampled from the set of {posterize, sharpness, solarize, equalize, brightness change, color change, auto-contrast}, leading to $I_{aug}$. Finally, $I_{aug}$ is masked by $M_a$ and blended with the original image I, leading to $I_{syn} = (1 - M_a) \odot I + (1 - \gamma)(M_a \odot I) + \gamma(M_a \odot I_{aug})$. The operator $\odot$ denotes element-wise multiplication and $\gamma$ denotes the opacity parameter that is uniformly sampled from $\gamma \in [0.2, 1.0]$. Figure 4 showcases exemplary synthetic images with the corresponding annotation mask.

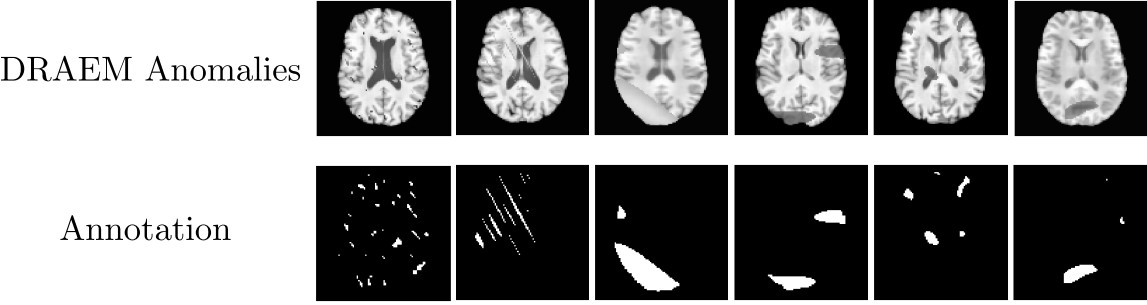

Figure 4: Examplary Synthetic Anomalies generated by the DRAEM procedure. Top: Images from the IXI data set, augmented with synthetic anomalies. Bottom: Annotation corresponding to the introduced anomalies.

## Appendix D. Analysis of the Anomaly Score Weighting

In this section, we analyze the different weightings of the anomaly scores from the supervised and reconstruction branches. We derive the AS by weighing the individual scores as follows

$$\text{Anomaly Score (AS)} = \beta \cdot x_{tilde} + (1 - \beta) \cdot \dot{\hat{y}}. \tag{4}$$

We vary the weighting parameter $\beta$ from zero to one. $\beta = 0$ corresponds to solely relying on the supervised branch ($\text{Unet}_{res}$). $\beta = 1$ corresponds to solely using the reconstruction branch (cDDPM).

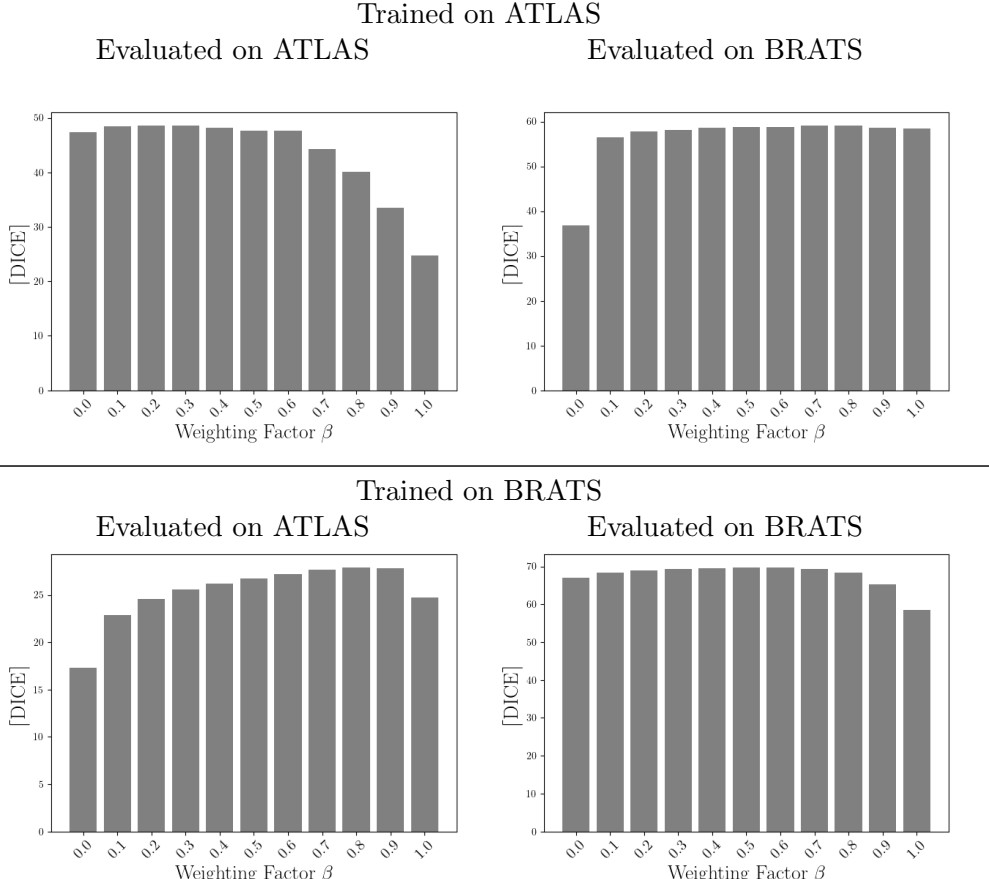

Figure 5: Analysis of the anomaly score weighting given
AS = $\beta \cdot x_{tilde} + (1 - \beta) \cdot \hat{y}$, where $x_{tilde}$ represents the anomaly map coming from the unsupervised reconstruction branch and $\hat{y}$ represents the anomaly map coming from the supervised segmentation branch. The ⌈DICE⌉ is plotted against different values of $\beta$.

