# OpenReview forum: "Combining Reconstruction-based Unsupervised Anomaly Detection with Supervised Segmentation for Brain MRIs"
_MIDL.io/2024/Conference — MIDL 2024 Oral_

### Official Review · Reviewer_ukmk · 2024-02-27

**Confidence:** 5
**Preliminary Rating:** 3
**Final Rating:** 4

**Summary:**

The authors propose a framework which combines reconstruction-based and self-supervised unsupervised anomaly detection. In the first training stage, pseudo-healthy reconstructions from an unsupervised diffusion model are learned. In the second stage, a segmentation network (Unet) is trained in a self-supervised utilising the generated pseudo-healthy images or in a semi-supervised way.

**Strengths:**

- Schematic drawing explains visually the proposed method
- Both qualitative and quantitative results are given
- Implementation details are given for all baseline models
- Synthetically generated anomalies are shown exemplarily in the appendix

**Weaknesses:**

- The authors test the proposed method on MRI Brain datasets, especially, training in an unsupervised or self-supervised (PII and DREAM) way on the IXI dataset. Unhealthy data is generated synthetically or taken from other datasets like ATLAS and BRATS. Brains are less variant than other body parts. A comparison, e.g., with Chest or Full-Body datasets would have been interesting and more convincing regarding the generalisability of the proposed method.

- The proposed method is evaluated against AE-based methods and Diffusion models. There was no comparison with SOTA Diffusion-model-based unsupervised/weakly supervised anomaly detection methods, e.g., [1,2].

- PII produces as in its original implementation only subtle synthetic anomalies which almost vanish in noise. The anomalies generated following Zavrtanik's DREAM framework are also less natural. A comparison with, e.g., naive hypo/hyper-intense synthetic anomalies would have been interesting. Also, a comparison with other SOTA Self-supervised methods, e.g., [3,4] would have been more convincing.

[1] Wolleb, J., Bieder, F., Sandkühler, R., & Cattin, P. C. (2022, September). Diffusion models for medical anomaly detection. In International Conference on Medical image computing and computer-assisted intervention (pp. 35-45). Cham: Springer Nature Switzerland
[2] Pinaya, W. H., Graham, M. S., Gray, R., Da Costa, P. F., Tudosiu, P. D., Wright, P., ... & Cardoso, M. J. (2022, September). Fast unsupervised brain anomaly detection and segmentation with diffusion models. In International Conference on Medical Image Computing and Computer-Assisted Intervention (pp. 705-714). Cham: Springer Nature Switzerland.
[3] Schlüter, H. M., Tan, J., Hou, B., & Kainz, B. (2022, October). Natural synthetic anomalies for self-supervised anomaly detection and localization. In European Conference on Computer Vision (pp. 474-489). Cham: Springer Nature Switzerland.
[4] Baugh, M., Tan, J., Müller, J. P., Dombrowski, M., Batten, J., & Kainz, B. (2023, October). Many tasks make light work: Learning to localise medical anomalies from multiple synthetic tasks. In International Conference on Medical Image Computing and Computer-Assisted Intervention (pp. 162-172). Cham: Springer Nature Switzerland.

**Detailed Comments:**

- Naming of models could be improved, especially, for 'SADM_{residual}' and 'SADM_{input}', and 'Final Anomaly Map'

**Justification Of Final Rating:**

The reviewer appreciates the authors' responsiveness to their feedback. With revisions addressing the reviewer's comments, the manuscript has shown improvement, leading the reviewer to upgrade their final rating to a 'weak accept.' In future submissions, the reviewer suggests expanding the evaluation beyond brain datasets to encompass less invariant data, as this could enhance the relevance and appeal of the method, e.g., regarding robustness.

**Justification Of The Preliminary Rating:**

The proposed method provides low contribution. The comparison with the aforementioned methods which are based on Diffusion-models/Pseudo-heathy images is missing. However, the method is easy to understand and to reproduce.

**Questions To Address In The Rebuttal:**

- This sentence is not fully clear: "Instead of learning the removal of synthetic anomalies during training, our model learns to remove any abnormal structure that is not part of the healthy training distribution." How do these learning strategies differ?

- Were there any experiments made with 3D instead of slice-wise training, and other imaging modalities?

- For post-processing, the authors apply median filtering with a kernel size of 5 to the generated anomaly maps. How does this influence the performance regarding tiny/small anomalies?

- How much training time was needed for training stage 1+2? Is it possible to merge both stages into one?

**Special Issue:**

No

---

> ### Author Response · Authors · 2024-03-17
> **Response to reviewer ukmk**
>
> Regarding the weaknesses:
>
> While we agree that considering other data sets and body regions would be interesting, we would like to note that the present study is already fairly comprehensive with respect to UAD in brain MRI. Furthermore, we feel that a broader scope would not fit this type of paper.
> Regarding the comparison with the state-of-the-art (SOTA), we believe that we provide a comprehensive assessment by comparing our method to three established approaches based on diffusion models, namely AnoDDPM, pDDPM, and cDDPM.
> The reviewer is right that the synthesized anomalies used in DRAEM are not very natural. However, we show that even with these rather unrealistic synthetic anomalies, our proposed framework is able to generalize to real pathologies, which is not the case for the plain self-supervised methods. We expect further performance improvements when using more natural and realistic anomalies in our framework and are curious to pursue this research direction in the future.
>
> Regarding the questions to address in the rebuttal
>
> 1.	This sentence is not fully clear: "Instead of learning the removal of synthetic anomalies during training, our model learns to remove any abnormal structure that is not part of the healthy training distribution." How do these learning strategies differ?
>
> We thank the reviewer for pointing this out and will clarify the statement in the following.
> The reconstruction in DRAEM training is enforced to replace the synthetic anomalies by calculating the reconstruction loss between the reconstructed image and the original input image without added anomalies. Therefore, the reconstruction model is directly trained to remove a certain type of anomalies. In contrast, we train our reconstruction model in an unsupervised fashion to reconstruct normal/healthy training data without explicitly enforcing the removal of certain anomalies.
> We adapted the description of DRAEM in section 2 to provide a clearer description in the revised manuscript:
> “Note that for the generator network in DRAEM, inpainting of synthetic anomalies is enforced by calculating the reconstruction loss between reconstruction and the anomaly-free input. In contrast, in our approach, the reconstruction model is trained on healthy data in an unsupervised fashion to remove any abnormal structure that is not part of the healthy training distribution.”
>
> 2.	Were there any experiments made with 3D instead of slice-wise training, and other imaging modalities?
>
> We conducted experiments to expand the underlying cDDPM reconstruction model to 3D. While observing some performance improvements, training the models in 3D is computationally demanding. Therefore, we focused on slice-wise processing for this study. However, we believe that our method holds promise for application in 3D settings. Also, while it would be interesting to evaluate the presented approach on other datasets and modalities, we had to focus on one scenario for the purpose of this work.
>
> 3.	For post-processing, the authors apply median filtering with a kernel size of 5 to the generated anomaly maps. How does this influence the performance regarding tiny/small anomalies?
>
> In general, larger kernel sizes would benefit the detection of larger anomalies while smaller anomalies could be filtered out. However, a kernel size of 5 is a common choice [1,2,3,4] that has not been shown to decrease the performance of smaller anomalies in our experiments.
>
> 4.	How much training time was needed for training stage 1+2? Is it possible to merge both stages into one?
>
> When training SADM_{residual} for synthetic anomalies, the training times for Stage I and II were approximately 5.5 and 2 hours, respectively, on an NVIDIA RTX 3090 GPU. Merging the training stages would be an interesting extension to our approach. We expect that training both stages in an end-to-end fashion is possible but might require a proper weighting of the individual losses.
>
> Minor comments:
>
> •	Naming of models could be improved, especially, for 'SADM_{residual}' and 'SADM_{input}', and 'Final Anomaly Map'
>
> Thank you for pointing this out. We changed ‘Final Anomaly Map’ to Anomaly Score (AS). Furthermore, we changed 'SADM_{residual}’ to ‘SADM_{res}’ and 'SADM_{input}’ to ‘SADM’. The same applies to UNET.
>
>
> [1] Baur et al., “Autoencoders for unsupervised anomaly segmentation in brain mr images: A comparative study.”,2021, Medical Image Analysis 69
>
> [2] Dey et al., “ASC-Net: Adversarial-Based Selective Network for Unsupervised Anomaly Segmentation”, 2021, MICCAI 2021
>
> [3] Akrami et al., “Deep quantile regression for uncertainty estimation in unsupervised and supervised lesion detection.”, 2022, MELBA (1 - IPMI Special Issue)
>
> [4] Kascenas et al., “Denoising Autoencoders for Unsupervised Anomaly Detection in Brain MRI”, 2022, MIDL 2022

---

### Official Review · Reviewer_V5u5 · 2024-02-28

**Confidence:** 5
**Preliminary Rating:** 5
**Recommendation:** Oral
**Final Rating:** 5

**Summary:**

In this paper, the authors build on a previously proposed framework for UAD in industrial image (DRAEM), which they modify in several  ways to produce a novel method, adapted to unsupervised anomaly detection (UAD) or weakly supervised anomaly detection. The authors design several experiments, performed on multiple public datasets, to prove the superiority of their proposed method.

**Strengths:**

1) Method is novel
2) Datasets used are public
3) The paper is very well written and easy to follow
4) The experiments conducted are numerous and relevant, with two testing sets, many baseline models, mutliple weakly supervised setups.
5) The rationale is clearly exposed and the choice of the method well motivated.
6) Every detail necessary to reproduce the experiments are provided
7) The differences between this paper and the main method used as inspiration (DRAEM) are clearly highlighted

**Weaknesses:**

1) The anomalies detected in both testing sets seem big and obvious to see, suggesting that a more challenging task would reinforce the findings of this study.
2) The reviewer believes both the noise used for the DDPM and the synthetic anomalies used for supervised segmentation are quite fitted (same statistics, size and intensity) to the anomalies tested for detection, which could hinder the performances for detection of other, different, types of anomalies.

**Detailed Comments:**

- "In stage II, the parameters ϕ are fixed" : the reviewer believe the authors meant \theta
- Can the authors precise the term "FAE" used to describe Meissen 2022b work ? At a quick glance the reviewer did not find correspondance to such an acronym in Meissen's work.

**Justification Of Final Rating:**

The reviewer thanks the authors for the comments and insightful discussions. The reviewer thinks the highlighted weaknesses were addressed by the authors and will keep his rating.
The reviewer still disagree with "self-supervised methods involve introducing synthetic anomalies and training a segmentation network to predict the anomaly's location", as the reviewer believes that self-supervised methods are much larger than this, even when restricted to UAD. However this is a very minor comment as the authors have precised what they meant in the manuscript.
Thanks again for the responses and discussion.

**Justification Of The Preliminary Rating:**

The reviewer is confident that this paper fits the conference expectations and is valuable to the field of UAD in medical imaging. Very detailed, relevant and argumented opinions of other reviewers could make the reviewer change his opinion.

**Questions To Address In The Rebuttal:**

- The reviewer believes the term "self-supervised", as used here, would have to be precised or changed. For instance using an auto-encoder reconstruction error (for training and testing) is considered self-supervised learning. The reviewer does not understand why the authors differentiate between self-supervised methods and reconstruction methods. Indeed the proposed method is fully unsupervised (or weakly-supervised), despite using supervised training, which can be difficult to grasp at a first reading, but this does not justify in the reviewer's point of view the use of "self-supervised" wording.
- Can the authors briefly comment on the use of l1 reconstruction loss instead of l2 ?
- Can the authors precise the window size used for SSIM ?
- Can the authors comment on why it is necessary to register the volumes to a common atlas ?
- The authors used the sum of the unsupervised anomaly map (x_tilde) and supervised anomaly prediction (y_dot_hat) as anomaly score map. The reviewer would be very curious to see an additional experiment, perhaps in appendix, of any metric, say Dice, plotted against a parameter $\Beta$, where the score map used would be  $\Beta$ x_tilde + (1-$\Beta$)y_dot_hat. Visualizing the peak and more generally the profile of such a curve would allow to see the importance and usefulness of each of the score maps used.
- The reviewer would like the authors to comment the fact that SADM_residual has rhoughly the same performances as SADM_input for the weakly supervised setting. Furthermore, the reviewer believes the sentence "Furthermore, feeding the residual of input and reconstruction to the Unet (SADMresidual )instead of the abnormal input only (SADMinput ) enhances the segmentation performance." has to be nuanced in light of the previous comment.
- Although common in UAD for medimage, the reviewer would appreciate if the authors can comment on the fact that the anomalies detected are still quite big and obvious to see (ATLAS and BraTS), and as a perspective refer to more challenging UAD tasks.

**Special Issue:**

Yes

---

> ### Author Response · Authors · 2024-03-17
> **Response to reviewer V5u5**
>
> 1.	The reviewer believes the term "self-supervised", as used here, would have to be precised or changed. For instance using an auto-encoder reconstruction error (for training and testing) is considered self-supervised learning. The reviewer does not understand why the authors differentiate between self-supervised methods and reconstruction methods. Indeed the proposed method is fully unsupervised (or weakly-supervised), despite using supervised training, which can be difficult to grasp at a first reading, but this does not justify in the reviewer's point of view the use of "self-supervised" wording.
>
> We agree with the reviewer that the terms “self-supervised” and “reconstruction-based methods” are overlapping. From our understanding of the UAD literature, there seems to be an established naming convention, though: Reconstruction-based approaches typically rely on the disparity between the input and its reconstruction for anomaly detection. On the other hand, self-supervised methods involve introducing synthetic anomalies and training a segmentation network to predict the anomaly's location. We clarified the naming in Section 2: “We note that while AE-based reconstruction methods may also fall under the category of self-supervised techniques, within this work, the term "self-supervised" refers to the aforementioned approach of training segmentation models using synthetic anomalies.”
>
> 2.	Can the authors briefly comment on the use of l1 reconstruction loss instead of l2 ?
>
> We use the l1 reconstruction error as this has shown improved reconstruction quality over the l2 error on the healthy IXI data set in our experiments.
>
> 3.	Can the authors precise the window size used for SSIM ?
>
> We calculate the SSIM anomaly score with a Gaussian kernel of size
> k_{ssim} = 9
> We added this information to the Manuscript.
>
> 4.	Can the authors comment on why it is necessary to register the volumes to a common atlas ?
>
> Registration of volumes to a reference atlas is a preprocessing step commonly applied in UAD. It aims to homogenize the data samples, ensuring they are consistently aligned to a spatial reference frame. While it may not be mandatory, especially in scenarios with ample training data, registration reduces the inter- and intra-data set variability of training and test data sets, which is assumed to simplify the reconstruction and anomaly scoring, particularly when evaluating on multiple different data sets.
>
> 5.	The authors used the sum of the unsupervised anomaly map (x_tilde) and supervised anomaly prediction (y_dot_hat) as anomaly score map. The reviewer would be very curious to see an additional experiment, perhaps in appendix, of any metric, say Dice, plotted against a parameter where the score map used would be BETA * x_tilde + (1-BETA)*y_dot_hat.
>
> We thank the reviewer for this insightful suggestion. We performed the suggested analysis in the Appendix, Figure 5. Overall, performance and generalization depend on the weighting factor BETA. Setting the beta to 0.5, as done in our experiments, provides a good initial choice and trade-off to generalize across in- and out-domain data. In further work, we plan to investigate more sophisticated mechanisms to combine the anomaly scores and, for instance, incorporate uncertainty estimates of the predictions.
>
> 6.	The reviewer would like the authors to comment the fact that SADM_residual has rhoughly the same performances as SADM_input for the weakly supervised setting. Furthermore, the reviewer believes the sentence "Furthermore, feeding the residual of input and reconstruction to the Unet (SADMresidual )instead of the abnormal input only (SADMinput ) enhances the segmentation performance." has to be nuanced in light of the previous comment.
>
> We acknowledge that performance improvements from SADM_{input} to SADM_{residual} are moderate in the weakly-supervised setting. In contrast, in the self-supervised setting, SADM_{input} substantially impedes the segmentation performance, compared to SADM_{residual}.
> We have revised the discussion: "Furthermore, feeding the residual of input and reconstruction to the Unet (SADMresidual ) instead of the abnormal input only (SADMinput ) can enhance the segmentation performance particularly in the self-supervised setting.”
>
> 7.	Although common in UAD for medimage, the reviewer would appreciate if the authors can comment on the fact that the anomalies detected are still quite big and obvious to see (ATLAS and BraTS), and as a perspective refer to more challenging UAD tasks.
>
> Generally, we agree with the reviewer that anomalies in BraTS and ATLAS are rather large and often fairly clear to identify. However, the latter is not equally true for all anomalies. We agree that extending our analysis to further datasets would be interesting.  We added a comment at the end of our manuscript: “Additionally, we intend to include data sets featuring subtler anomalies or different imaging modalities to broaden the evaluation of our approach.”.

---

> > ### Author Response · Authors · 2024-03-17
> > **Response to reviewer V5u5**
> >
> > Detailed Comments:
> >
> > 1.	"In stage II, the parameters ϕ are fixed" : the reviewer believe the authors meant \theta
> >
> > We thank the reviewer for identifying this and revised manuscript accordingly.
> >
> > 2.	Can the authors precise the term "FAE" used to describe Meissen 2022b work ? At a quick glance the reviewer did not find correspondance to such an acronym in Meissen's work.
> >
> > The reviewer is right. There is no such term in the cited work of Meissen 2022b. Unfortunately, we referred to a different work of the same authors. The right reference would be:
> > [Meissen, F., Paetzold, J., Kaissis, G., Rueckert, D., 2022b. Unsupervised anomaly localization with structural feature-autoencoders. arXiv preprint arXiv:2208.10992].
> > We apologize for the confusion. We corrected the reference in the revised manuscript.

---

> > ### Comment · Reviewer_V5u5 · 2024-03-24
> >
> > The reviewer thanks the authors for the comments and insightful discussions. The reviewer thinks the highlighted weaknesses were addressed by the authors and will keep his rating.
> > The reviewer still disagree with "self-supervised methods involve introducing synthetic anomalies and training a segmentation network to predict the anomaly's location", as the reviewer believes that self-supervised methods are much larger than this, even when restricted to UAD. However this is a very minor comment as the authors have precised what they meant in the manuscript.
> > Thanks again for the responses and discussion.

---

### Official Review · Reviewer_sxA9 · 2024-02-29

**Confidence:** 3
**Preliminary Rating:** 4
**Final Rating:** 4

**Summary:**

The paper introduces a novel approach that integrates reconstruction-based and self-supervised anomaly detection techniques for brain MRIs. Initially, the method trains a diffusion model to reconstruct healthy brain images. Subsequently, these reconstructions are leveraged to train a Unet in a supervised manner for precise anomaly segmentation. Finally, by integrating both networks, the approach generates anomaly maps for test images. Through experimental evaluation, the method demonstrates advantages in detecting not only in-domain anomalies but also new unseen ones when compared to baselines.

**Strengths:**

- The paper is well-written and self-contained, providing a detailed and clear description of the methodology employed.
- It proposes an interesting approach by incorporating unsupervised diffusion models with supervised segmentation models for the task of anomaly detection.
- Through proper experimental comparisons with baseline methods, the authors effectively demonstrate the improved performance of their proposed method and its variants.
- A significant advantage of the proposed method lies in its capability to address various training settings i.e., weakly-supervised schemes, by leveraging available annotations.

**Weaknesses:**

- While not an actual weakness, it would be helpful to include an evaluation of a fully-supervised setting, such as training a supervised Unet on a larger subset of the BRATS/ATLAS/DRAEM datasets and then evaluating it on the rest of them. This comparison would allow for an assessment against the presented methods designed for minimal annotation settings, quantifying their gap to the supervised scenario.

**Detailed Comments:**

- Considering that brain images are processed slice by slice, is there a bottleneck when iterating through all slices within a test image volume to generate its segmentation map?
- In Table 1, it appears that the proposed method underperforms in blocks 2 and 3 in the DRAEM test dataset when compared to the baselines. Could you confirm this? Additionally, do you have any intuition for this?
- There is a typo on page 3/Figure 1: “In stage II, the parameters ϕ are fixed,” where ϕ should be θ.

**Justification Of Final Rating:**

The reviewer appreciates the authors' feedback and their additional efforts to refine their manuscript. They note that minor points have been clarified, contributing to the paper's overall coherence. The proposed methodology is interesting and well-presented. The reviewer thanks the authors and believes their contribution will be valuable for the conference.

**Justification Of The Preliminary Rating:**

The paper introduces an interesting approach, which is then presented in a clear and well-documented way. Through extensive experimentation, the authors systematically compare their method against baselines in various settings, effectively showcasing its improved performance and enhanced generalizability.

**Questions To Address In The Rebuttal:**

The paper is well-organized, justifying design choices through intuition and experimental results. A brief consideration of the minor suggestions outlined above could add to its completeness.

---

> ### Author Response · Authors · 2024-03-17
> **Response to reviewer sxA9**
>
> 1.	Considering that brain images are processed slice by slice, is there a bottleneck when iterating through all slices within a test image volume to generate its segmentation map?
>
> The reviewer is right that sequential slice-wise processing of volumes can slow down the reconstruction process compared to 3D processing. However, we implement the reconstruction in a parallel fashion to reconstruct all slices simultaneously. This reduces the bottleneck, and the reconstruction of a full volume takes approximately 0.5 seconds on an NVIDIA RTX 3090 GPU.
>
> 2.	In Table 1, it appears that the proposed method underperforms in blocks 2 and 3 in the DRAEM test dataset when compared to the baselines. Could you confirm this? Additionally, do you have any intuition for this?
>
> Indeed, regarding synthetic anomalies, our proposed method is outperformed by the fully supervised Unet baselines. This may indicate, that the supervised methods focus on clear on distinct features when detecting anomalies and that the synthetic data show rather distinct differences. Also, the fact that the supervised methods are not superior when applied to the BraTS and ATLAS datasets supports this hypothesis, i.e., the real pathologies may show more subtle patterns.
>
> 3.	There is a typo on page 3/Figure 1: “In stage II, the parameters ϕ are fixed,” where ϕ should be θ.
>
> Thank you for pointing this out. We corrected the typo in the revised manuscript.

---

### Author Response · Authors · 2024-03-17
**Response to all reviewers**

Dear reviewers, thank you for the detailed reviews and constructive feedback on our work. We are delighted to see that the reviewers appreciated the experimental design, novelty, and contributions our work offers to the field of UAD in brain MRI. Additionally, we are pleased to acknowledge the recognition of our efforts to ensure the reproducibility, clarity and replicability of our research.
We also recognize the identified directions to improve our work. Specifically, questions have been raised regarding the generalizability of our approach to different datasets, pathologies, and imaging modalities. We will thoroughly consider your feedback and answer the questions you raised. Additionally, we provide a revised manuscript that addresses the feedback and includes additional experiments. Changes in the revised manuscript are color-coded in red.
Once again, we express our gratitude for the thorough evaluation of our work and the constructive and detailed suggestions provided. We strongly believe that your feedback allowed us to enrich the quality of our manuscript and provided us the opportunity to steer the direction of our future.

---

### Meta-Review · Area_Chair_cRUa · 2024-04-03

**Recommendation:** Accept (Poster)
**Confidence:** 5

**Metareview:**

The Meta-Reviewer has read all reviews and rebuttal of authors.

Reviewers have all confirmed they have taken the author rebuttal into account for their final recommendation.

There is a consensus among reviewers that the work is at the level required for publication and would make a positive addition to the conference.

---

### Decision · Program_Chairs · 2024-04-05

Accept (Oral)